# Ethylene, ACC, and the Plant Growth-Promoting Enzyme ACC Deaminase

**DOI:** 10.3390/biology12081043

**Published:** 2023-07-25

**Authors:** Elisa Gamalero, Guido Lingua, Bernard R. Glick

**Affiliations:** 1Dipartimento di Scienze e Innovazione Tecnologica, Università del Piemonte Orientale, Viale T. Michel 11, 15121 Alessandria, Italy; guido.lingua@uniupo.it; 2Department of Biology, University of Waterloo, Waterloo, ON N2L 3G1, Canada; glick@uwaterloo.ca

**Keywords:** ethylene, 1-aminocyclopropane1-carboxylate, ACC deaminase, D-cysteine desulfhydrase, D-amino acids, plant evolution

## Abstract

**Simple Summary:**

The molecule 1-aminocyclopropane-1-carboxylate is the immediate precursor of the plant hormone ethylene in most seed plant species. Both 1-aminocyclopropane-1-carboxylate and ethylene can affect plant growth and development in a variety of ways. In addition, the bacterial enzyme 1-aminocyclopropane-1-carboxylate deaminase can cleave 1-aminocyclopropane-1-carboxylate and prevent both 1-aminocyclopropane-1-carboxylate and ethylene from affecting plant gene expression and subsequent behavior. In this review, the roles of 1-aminocyclopropane-1-carboxylate, ethylene and 1-aminocyclopropane-1-carboxylate deaminase in the development of plants and their regulation are explored. Since D-amino acids stimulate ethylene synthesis in plants, a section of the review is dedicated to this topic. It is suggested that 1-aminocyclopropane-1-carboxylate was synthesized in plants prior to the plant synthesis of ethylene, giving 1-aminocyclopropane-1-carboxylate some control of plant gene expression in response to environmental signals.

**Abstract:**

Here, a brief summary of the biosynthesis of 1-aminocyclopropane-1-carboxylate (ACC) and ethylene in plants, as well as overviews of how ACC and ethylene act as signaling molecules in plants, is presented. Next, how the bacterial enzyme ACC deaminase cleaves plant-produced ACC and thereby decreases or prevents the ethylene or ACC modulation of plant gene expression is considered. A detailed model of ACC deaminase functioning, including the role of indoleacetic acid (IAA), is presented. Given that ACC is a signaling molecule under some circumstances, this suggests that ACC, which appears to have evolved prior to ethylene, may have been a major signaling molecule in primitive plants prior to the evolution of ethylene and ethylene signaling. Due to their involvement in stimulating ethylene production, the role of D-amino acids in plants is then considered. The enzyme D-cysteine desulfhydrase, which is structurally very similar to ACC deaminase, is briefly discussed and the possibility that ACC deaminase arose as a variant of D-cysteine desulfhydrase is suggested.

## 1. Introduction

Plant growth and development are regulated by several different phytohormones including cytokinins, gibberellins, auxins, salicylic acid, jasmonates, brassinosteroids, abscisic acid, strigolactones, and ethylene [1,2,3,4,5,6]. A key phytohormone, and of particular interest to this manuscript, is ethylene (C_2_H_4_), a low molecular weight (28.05 g/mol) gaseous hydrocarbon that is produced in all higher plants and modulates a wide range of plant physiological and biochemical activities [7,8,9]. Thus, for example, “ethylene is involved in seed germination, tissue differentiation, formation of root and shoot primordia, root branching and elongation, lateral bud development, flowering initiation, anthocyanin synthesis, flower opening and senescence, fruit ripening and degreening, production of volatile organic compounds, …aroma formation in fruits, storage product hydrolysis, leaf senescence, leaf and fruit abscission, Rhizobia nodule formation, mycorrhizae-plant interaction, and (importantly) the response of plants to various biotic and abiotic stress” [7]. The impact of ethylene on a particular plant trait may be either stimulatory or inhibitory, and is a consequence of the genus and species of the plant, the age of the plant, and the soil conditions, including the presence of soil microbes, the weather, and the amount of ethylene that is produced. Moreover, depending upon these conditions, a very wide range of ethylene concentrations (~4000-fold) may exhibit biological activity [7,10]. In addition, in some plants under specific conditions, the presence of increasing ethylene may impact the synthesis of other phytohormones including abscisic acid, gibberellin, cytokinin, and auxin [11,12]. Thus, some of the biological activities largely attributed to ethylene may be the consequence of ethylene affecting the concentration of, or acting in concert with, other phytohormones.

In this review, the role of the phytohormone ethylene and 1-aminocyclopropane-1-carboxylate (ACC), ethylene’s immediate precursor, to act as signaling molecules in plants is reviewed and discussed. In addition, the evidence for the ability of the widespread soil microbial enzyme ACC deaminase [13] to modulate ethylene [14,15,16] and ACC [17] signaling in primitive as well as modern plants is discussed. The similarity between ACC deaminase and the enzyme D-cysteine desulfhydratase [13,18] is also discussed and the possibility that ACC deaminase originated as a variant form of D-cysteine desulfhydratase is considered. Finally, the possible role of D-cysteine desulfhydratase and D-amino acids in general in primitive plants is addressed.

## 2. Ethylene and ACC Biosynthesis in Plants

The biosynthesis of ethylene has been studied in detail in numerous higher plants and it appears that all of these plants utilize essentially the same mechanism to synthesize ethylene [19,20]. In addition to the plant biosynthesis of ethylene, some microorganisms can also synthesize ethylene, albeit using an entirely different biosynthetic pathway [21,22]. In plants, the biosynthesis of ethylene begins with the conversion of the relatively rare, but nonetheless very important, amino acid L-methionine into the compound *S*-adenosylmethionine (SAM) by the enzyme SAM synthase, which is encoded by a small multi-gene family (Figure 1). 

Being at the junction of several biosynthetic pathways [23], SAM is moderately abundant within plant tissues. For the synthesis of ethylene, the compound SAM is converted into 1-aminocyclopropane-1-carboxylate (ACC) and 5′-methylthioadenosine (MTA) by the enzyme ACC synthase [24,25]. The MTA that is formed as a byproduct of this reaction is recycled to form the amino acid L-methionine. This allows the amount of L-methionine in a plant cell to remain relatively constant even during fairly high rates of ethylene production. Some scientists believe that the synthesis of ACC from SAM is the committal, or rate-limiting, step in the biosynthesis of ethylene [26,27]. There are several, nearly identical, ACC synthase enzymes present in a plant cell as a consequence of the fact that the genes that encode this enzyme are part of a multi-gene family. Moreover, considerable evidence suggests that the transcription of different genes that encode this enzyme are regulated under a range of different environmental or plant physiological conditions (e.g., [25]). When high levels of ACC synthesis are not required, the amount of ACC synthase in plant cells remains relatively low. The subsequent conversion of ACC to ethylene is catalyzed by the enzyme ACC oxidase [28,29,30], which is also generally present constitutively in most plant tissues at very low levels. Similar to ACC synthase genes, ACC oxidase genes are part of a multi-gene family with different isoforms of this enzyme being actively transcribed under different environmental or plant physiological conditions [31]. Thus, in many physiological conditions, both ACC synthase and ACC oxidase may be considered to be inducible enzymes. The fact that both enzymes are inducible is somewhat unusual in that in most metabolic pathways that have been studied only a single step is typically thought of as rate limiting. However, inducible ACC synthesis followed by inducible ethylene synthesis is consistent with the idea, posited below, that ACC synthesis and ethylene synthesis evolved separately from one another.

Plant cells often make more ACC than they require at any particular time. This enables them to rapidly respond to changing environmental conditions and to quickly synthesize ethylene from this storehouse of ACC. However, to remove some of the excessive ACC when it is not immediately needed, plant cells are able to convert ACC into inactive conjugated forms of this compound. For example, the conjugation of ACC with malonate or glutathione occurs as a consequence of the action of either the enzyme ACC N-malonyl transferase [32,33] or the enzyme gamma-glutamyl transpeptidase [34]. These reactions result in the production of either 1-(malonyl)-ACC (M-ACC) or 1-(glutamyl)-ACC (G-ACC), respectively (Figure 1). Kinetic studies have determined that the tightness of the enzyme ACC N-malonyl transferase binding to its substrate ACC is much lower than the tightness of the enzyme ACC oxidase for the same substrate. Thus, if these two enzymes are present in cells in similar amounts, the ACC will bind preferentially to ACC oxidase, and it will subsequently be converted into ethylene [19]. However, as indicated earlier, there is often an excess of ACC in the cell that needs to be removed, and it would be detrimental to the plant to convert all the available ACC into ethylene [35]. In this regard, the conjugation reactions remove ACC only when it is present in relatively high levels, i.e., there is more ACC than is required for the necessary ethylene synthesis. 

## 3. Ethylene as a Signaling Molecule

Ethylene is one of the simplest signaling molecules with hormone-like behavior that is synthesized by plants. When it plays the role of a hormone, it regulates plant development (seed germination, cell elongation, fruit ripening, seed dispersal) and plant responses to environmental stresses such as soil pollution by metals, high salinity levels, low water availability, and sub-optimal temperatures, as well as pathogen (fungi, insects, nematodes) attack. The long history of research on plant ethylene, well described in the review paper written by Bakshi et al. [36], started in 1885 with the first observation by George Fahnestock reporting that illuminating gas used for lighting in both homes and streets negatively affected plant health and growth in a greenhouse in Philadelphia [37]. About fifteen years later, Dimitry Neljubow, a plant physiologist at the Botanical Institute of St. Petersburg University in Russia, identified ethylene as the active molecule in illuminating gas that affects plants [38]. Since then, a quite large number of papers have focused on the roles of ethylene in plants and on its biosynthetic pathways (see Section 2 of this review), and only more recently has attention been given to the perception of this gas through ethylene-binding sites and to the genes and proteins differentially expressed as a consequence of ethylene synthesis. All the major molecular elements involved in ethylene signaling pathways have been identified and described through a combination of molecular biology, cell biology, biochemistry, and genetic tools [10].

Based on the results of several studies, performed mainly on *Arabidopsis*
*thaliana*, it has been hypothesized that the plant gene(s) dedicated to ethylene perception are derived from a cyanobacterium that transferred this coding DNA to the chloroplast genome [39,40,41]. According to one pioneering study, it has been estimated that about 4000 ethylene-binding sites are distributed through each tobacco leaf cell [42], mainly located in the endoplasmic reticulum membrane [43,44]. All of the ethylene receptors described in the literature show a hydrophobic N-terminal domain comprising the ethylene-binding domain [41,45,46], followed by a cytosolic domain, which contains ubiquitous sequences occurring in a plethora of other signaling molecules expressed by members of all the three kingdoms of life, playing a pivotal role in protein–protein interactions between the receptors [47,48,49,50]. At the base of the ethylene receptor is a protein homodimer that binds noncovalently with other homodimers leading to the formation of higher order homomeric and heteromeric complexes [49].

The binding of ethylene with its receptor is supported by a copper-based cofactor, which is required for ethylene-receptor functions, and is provided by the RAN1 (Responsive to Antagonist1) copper transporter [41]. According to the studies performed on *A. thaliana*, there are five ethylene receptors in this model plant that can be categorized into two clades, the first one containing ETR1 and ESR1 (Ethylene Response 1 and Ethylene Response Sensor 1, respectively) and the second represented by ETR2, ERS2, and EIN4 (Ethylene Response 2, Ethylene Response Sensor 2, and Ethylene Insensitive 4, respectively) [51]. A structural model of this ethylene receptor has been proposed by Schott-Verdugo et al. [52] and, very recently, the structure of the ethylene-binding domain of ETR1 has been elucidated by Azhar et al. [53]. Based on the fact that the functioning of the ethylene receptor is dependent on copper availability, it has been hypothesized that the whole mechanism of ethylene perception depends on an ancient copper transport mechanism that protects plant cells from the toxicity induced by high concentrations of this metal [54]. However, since the publication of this paper, there have been no further findings in support of this hypothesis.

The gene *CTR1* (CONSTITUTIVE TRIPLE RESPONSE 1), coding for a serine/threonine protein kinase, behaves as a negative regulator of ethylene responses, where ethylene response in plants is suppressed by its protein kinase activity [55]. The N-terminal regulatory domain is closely connected to ETR1. Although this association is required in order to ensure the kinase activity of CTR1, the mechanism by which CTR1 is activated by the ethylene receptors is still unknown [56,57,58,59]. 

The protein EIN2 (ETHYLENE-INSENSITIVE 2), located in the endoplasmic reticulum membrane, shows 12 trans membrane domains at its N terminus and a plant specific domain that is involved in the activation of the ethylene downstream response at the C terminus [60,61]. It has been shown that *A. thaliana* has a single-copy EIN2 gene whose sequence is conserved from the charophyte green algae to land plants [62]. Its role is to transfer the ethylene plant response to EIN3 (ETHYLENE-INSENSITIVE 3), which is in the nucleus; in fact, when ethylene is perceived by the plant, EIN2 is cleaved by an unknown protease at the C terminal portion and the remaining sequence can move to the nucleus [63,64]. 

Additionally, EIN2-C can bind to the EBF1/EBF2 RNAs and be sequestered in cytoplasmic granules composed of translationally repressed mRNAs and proteins related to mRNA decay, called processing bodies (P-bodies) [16,64]. Finally, EIN2 stabilizes the two transcriptional factors EIN3 and EIL1 (ETHYLENE-INSENSITIVE 3-like 1 protein, homologous to EIN3), proteins which are key transcriptional factors involved in the modulation of the ethylene response genes such as ETHYLENE RESPONSE FACTORS (ERFs) [65,66]. The identification of this pathway allowed scientists to describe how an ethylene signal goes from the site of perception at the ER membrane and then to the nucleus. The pathway of ethylene signaling is depicted in Figure 2.

If ethylene is present (case B), it binds to the receptor and inactivates CTR1. This inactivation promotes the cleavage of the C terminus of EIN 2 protein. 

The EIN2 C-terminal domain (EIN2-C), released upon cleavage, inhibits the translation of EBF1/EBF2 thus allowing accumulation of the EIN3- and EIN3-LIKE1 (EIL1)-transcription factors that activate the transcription of ERF1 (ETHYLENE RESPONSE FACTOR 1) and of many other genes involved in ethylene response. Altogether, the transcription of these genes then activates the plant ethylene response. Furthermore, EIN2-C can bind to the EBF1/EBF2 RNAs and become sequestered in processing bodies (P-bodies) in the cytoplasm.

The response of plants to ethylene is the same regardless of whether the ethylene is exogenously provided or endogenously synthesized.

## 4. ACC as a Signaling Molecule

Several studies have provided evidence that, in addition to acting as a precursor for the synthesis of ethylene, ACC itself can act as a hormonal signal [7,17,27,66,67,68,69,70,71]. The first indication of this possibility, using chemical inhibitors of ethylene biosynthesis or ethylene perception, was that ACC appeared to be a signaling molecule for root to shoot communication under conditions where ethylene perception was blocked [66]. In other experiments, ACC was shown to play a role in stomal development in *A. thaliana* [72]. In addition, other studies suggested that ACC plays a direct role in plant defenses against the fungal phytopathogen *Verticillium dahliae* [73]. For example, during periods of flooding, ACC, which is primarily synthesized in plant roots, is transported through the xylem to the shoots where, as a consequence of the availability of oxygen, the ACC is converted to ethylene [74,75]. It is possible that the precise role of ACC as a signaling molecule might be better defined, at least in some cases, by repeating some of the experiments that have previously been performed using chemical ethylene (synthesis or perception) inhibitors or ethylene biosynthesis mutants and instead include the presence of the enzyme ACC deaminase, which is an ACC rather than an ethylene inhibitor (see Section 5). Interestingly, it has been demonstrated that ACC behaves as a signal molecule involved in the recruitment of specific bacteria able to cleave ACC into ammonia and α-ketobutyrate, so that it shapes the rhizosphere microbiome. In turn, these bacteria reduce the stress levels in plants and this new physiological condition can subsequently modulate the composition of the plant-associated bacterial communities [76,77].

Finally, assuming that ACC is in fact a plant-signaling molecule, at least under certain circumstances, this is consistent with the possibility that ACC may have been a major signaling molecule in primitive plants prior to the development of ethylene and ethylene signaling, and prior to ACC becoming a precursor for the synthesis of ethylene in seed plants [78].

## 5. ACC Deaminase

Plant growth-promoting bacteria (PGPB) that contain the enzyme ACC deaminase provide a significant advantage to plants growing in nature, in that this enzyme cleaves ACC into ammonia and α-ketobutyrate (Figure 3) thereby preventing the ACC from being converted to ethylene. 

This is especially important during stressful conditions when plants often synthesize stress ethylene, which is usually inhibitory to plant growth [79,80]. For example, when scientists isolated PGPB from the rhizosphere of wild barley plants growing on two opposite slopes of a canyon in northern Israel, separated by ~250 m, they observed that the south- and north-facing slopes contained very different PGPB [81]. The south-facing slope was quite arid with a large amount of sunlight and had only very sparse plant growth. On the north-facing slope, the plant growth was quite lush. Both the south- and north-facing slopes contained similar plants (i.e., wild barley) and similar genera of bacteria within the plant rhizospheres. On closer examination, rhizosphere bacteria from the stressed south-facing slope all contained ACC deaminase, an activity that was present to a much lesser extent in the bacteria from the north-facing slope. Thus, the highly stressful conditions (drought and excess sunlight) on the south-facing slope selected for rhizospheric PGPB, containing ACC deaminase, which better enabled both the PGPB and the plants to survive under these harsh conditions. Importantly, ACC deaminase activity is not limited to the PGPB that occupy the rhizospheres of obviously stressed plants. Rather, it has been found in α, β, and γ Proteobacteria, Actinobacteria, Firmicutes, Bacteroidetes, Archaea, various fungi, and yeast [13,82].

Notwithstanding the abovementioned experiment, most studies of the efficacy of ACC deaminase in thwarting abiotic and biotic stresses have been performed in laboratory experiments, including both growth chamber and greenhouse experiments [83]. Based on many hundreds of reports of ACC deaminase containing PGPB protecting plants against a wide range of both abiotic and biotic stresses [84,85,86,87,88,89,90,91,92,93,94,95,96,97,98,99,100,101,102,103,104,105,106,107,108,109,110,111,112], it is clear that this enzyme is a key component of the ability of plants to survive and thrive in a multitude of different stressful environments.

Where it has been studied, ACC deaminase has been found to be a cytoplasmically localized enzyme that is not secreted to the external medium [113]. Rather, the substrate, ACC, must be taken up by the bacterium that contains this enzyme. The active form of ACC deaminase is multimeric (probably tetrameric) with one mole of the co-factor pyridoxal phosphate per mole of the monomeric enzyme subunit [114]. ACC deaminase enzymes from different microbe’s function optimally between pH 8.0 and 9.0 and typically have a subunit molecular mass of 33–42 kDa [115]. Analyses of ACC deaminase enzyme structures suggest that this enzyme is relatively thermodynamically stable with a Tm = ~60 °C [116].

A model that was developed to understand the promotion of plant growth and development by ACC deaminase from PGPB [15,117] includes several different steps (Figure 4). 

In this model, following the binding of a bacterium to a plant tissue (usually plant roots or plant seeds) or, in the case of endophytic bacteria, the uptake of the bacteria into plant tissues, the bacteria take up the amino acid L-tryptophan from the plant and use it to synthesize the phytohormone indoleacetic acid [118,119]. Some of the newly synthesized IAA is secreted by the bacteria (which are typically either bound to the surface of plant roots or localized within the root endosphere), taken up by plant tissue, and added to the endogenous IAA pool in the plant. IAA, through a series of metabolic steps (including auxin response factors and auxin transport proteins) [120], can either promote plant-cell growth and proliferation or activate the transcription of ACC synthase [121]. Increased ACC synthase transcription will result in an increase in ACC and ethylene synthesis with the increased amount of ethylene feed-back inhibiting auxin response factors [120] thereby limiting the additional functioning of IAA. Increasing IAA also results in plant-cell wall loosening [122] therefore increasing the amount of exudation of small molecules, including ACC, from the plant [84]. The exuded ACC is taken up by the plant-associated bacteria and cleaved into ammonia and alpha-ketobutyrate, both of which can be metabolized by the bacteria. This results in a decreased level of ethylene inside of the plant cells and a decreased amount of ethylene-caused feed-back inhibition of auxin response factors. Thus, in the presence of ACC deaminase, more IAA is directed toward plant-cell growth and proliferation. The net result of this scheme is that a PGPB that contains both ACC deaminase and the ability to synthesize IAA can (i) lower the ethylene inhibition of plant growth and (ii) simultaneously increase the IAA metabolic flux in the plant, both of which promote plant growth. It should be noted that while not all rhizospheric and endophytic PGPB produce IAA, multiple studies have suggested that around 85–90% of these bacteria have the ability to synthesize IAA. Finally, since both abiotic and biotic stresses generally increase plant ethylene levels, this model explains how ACC deaminase can decrease some of the deleterious effects of these stresses on plants.

## 6. D-Amino Acids in Plants

The L-form of amino acids, called proteinogenic amino acids, is by far the more prevalent one. There is a consensus on the fact that the prevalence of the L isoforms of amino acids dates back to ancient times. In fact, it has been reported that L-enantiomers of amino acids are dominant in the composition of a carbonaceous chondrite meteorite (Murchison meteorite) that fell in Australia in 1969 [123,124,125]. Since the formation of carbonaceous chondrites dates to ~4.5 billion years ago, it has been hypothesized that the amino acids’ molecular asymmetry appeared before the emergence of life on Earth [123]. This trait was probably randomly selected during the molecular evolution of life. Although amino acid homochirality is a quite rigorous condition, proteins and peptides containing D-amino acids are synthesized mainly after the incorporation of D-amino acids by a non-ribosomal peptide synthetase (NRPS) or through post translational modifications of peptide precursors based on L-amino acids [126]. While the first strategy is followed only by prokaryotes, the second one is exploited by both prokaryotes and eukaryotes. When the first occurrence of D-amino acids was observed in organisms, it was thought that they were peculiar but without any specific role. More recently, it has been demonstrated that D-amino acids have several structural and physiological functions. For example, the D-amino acids, D-alanine and D-glutamate, together with other D-amino acids such as D-aspartate and D-serine, are fundamental building blocks in bacterial peptidoglycan synthesis [127]. These molecules are generally considered to be detrimental for plant growth; however, plants are able to synthesize D-amino acids thus suggesting a functional role for these molecules in plants [128]. In fact, it has been demonstrated that D-serine behaves as a signal molecule modulating pollen growth in the pistil of *A. thaliana* [129].

Plants are constantly in contact with D-amino acids, which are very concentrated in the soil (in the order of milligrams per kg of soil) surrounding the root system [130]. Once they are transported inside plants, D-amino acids may be used as a nitrogen source modulating chloroplast division and the production of ethylene and affecting plant development according to the D-amino acid and the plant species involved [131,132,133]. Moreover, several genes involved in D-amino acid synthesis have been detected in plant genomes [134].

Regarding D-amino acid transport inside plants, workers have identified a lysine histidine transporter 1 (LTH1) as being responsible for the uptake of different amino acids at the root level [131,135]. Besides LTH1, two other families of transporters have been found to be able to take up D-amino acids. They are (1) the Amino acid Permease 1 (AAP1), which is mainly responsible for the uptake of D-methionine and D-phenylalanine [136], and (2) the proline transporter family ProT [137]. The common trait among these uptake systems is that they are not specific for D-amino acids since they are primarily responsible for the transport of L-amino acids.

On the other hand, D-amino acids are passively exuded through diffusion in *A. thaliana* [138]. This is interesting, since the uptake of D-amino acids occurs through an active ATP-consuming mechanism while their release by roots is mediated using passive transport [138]. Whether D-amino acids, together with all the other molecules contained in rhizo-deposits, are involved in the stimulation, inhibition, or establishment of certain bacterial communities remains to be elucidated. 

As previously stated, according to the plant species and the D-amino acid considered, the effect of these molecules on plant development varies. The exogenous treatment of pepper plants with a mixture of D-Leucine, D-valine, and D-cysteine led to growth promotion possibly related to the utilization of these amino acids as a nitrogen source [130]. On the other hand, negative effects on plant growth were induced in *A. thaliana* by treatments with D-alanine at concentrations higher than 0.5 mM [131]. 

In some plant species, exogenous treatment with D-methionine enhanced the ethylene level. Interestingly, in *A. thaliana*, D-methionine is one of the preferred substrates of the enzyme AtDAT1 (a transaminase specific for D-amino acids). In the presence of D-methionine, the seeds of *A. thaliana* that lack the gene encoding AtDAT1 or are unable to synthesize this enzyme show a decreased level of germination due to increased ethylene levels. In fact, mutants lacking this gene are characterized by a high malonylation of D-methionine, combined with a reduction in malonyl-ACC content, which is the major product of ACC degradation [139]. Consequently, D-methionine regulates ethylene synthesis with high concentrations of D-methionine leading to this D-amino acid to outcompete ACC for the enzyme N-malonyl-transferase. As a result, the high amount of residual ACC is oxidized to ethylene by the enzyme ACC oxidase [30].

A functional role has also been recognized for D-cysteine, behaving as a precursor of hydrogen sulfide that is involved in plant stress responses [140]. Finally, a study performed on the moss *Physcomitrella patens* (now *Physcomitrium patens*) revealed the presence of the dipeptide D-Ala-D-Ala, as it typically occurs in the bacterial peptidoglycan, in the plastid envelope. Mutants of *P. patens* lacking the ability to synthesize the D-Ala-D-Ala dimer are hampered in plastid division [141]. Since the mutants of *A. thaliana* that lack the orthologous genes remained unaffected in plastid division, it has been hypothesized that the ability to synthesize peptidoglycan and integrate it into plastidic envelopes is a trait that has been lost during the evolution of land plants after the development of lycophytes [142].

However, it is also true that genes for peptidoglycan synthesis have been found in *Picea abies* and *Pinus taeda* and at least four genes (if not the complete set) involved in peptidoglycan formation have been detected in several angiosperms [143]. Thus, the possible presence of peptidoglycan in higher plants is still sparking scientific debate.

## 7. D-Cysteine Desulfhydrase

D-cysteine desulfhydrase is a pyridoxal-5′-phosphate-dependent enzyme that breaks down D-cysteine into pyruvate, hydrogen sulfide, and ammonia [144]. This enzyme can also break down β-chloro-D-alanine [145]. This enzyme has been detected in both plants [18,146,147,148,149] and bacteria [144,150]. Moreover, the amino acid sequence of D-cysteine desulfhydrase is highly homologous to several ACC deaminases [13,18,150] and the ACC molecule is able to bind to the active site of crystallized D-cysteine desulfhydrase from *Salmonella typhimurium* [145] even though D-cysteine desulfhydrase does not have any ACC deaminase activity. In *E. coli* and yeast, D-cysteine is toxic to the microbes, possibly because of its inhibition of the activity of the enzyme threonine deaminase [150]. In addition to plants and microbes, D-cysteine desulfhydrase activity has been purified and characterized from the green alga *Chlorella fusca* [147]. Interestingly, the properties of all the investigated D-cysteine desulfhydrases are quite similar. 

In a key experiment, tomato (*Solanum lycopersicum*) cDNA that was predicted to encode a putative ACC deaminase was isolated and expressed in *E. coli* [18]. However, a detailed kinetic assessment of this expressed cDNA revealed that it encoded D-cysteine desulfhydrase and not ACC deaminase. The subsequent site-directed mutagenesis of this cDNA, altering the codons for two amino acid residues that were within the predicted active site of the enzyme D-cysteine desulfhydrase, changed the activity of the enzyme from D-cysteine desulfhydrase to ACC deaminase. Moreover, the site-directed mutagenesis of the codons for two amino acid residues within the same position of the active site of ACC deaminase from the plant growth-promoting bacterium *Pseudomonas* sp. UW4 [151] changed its activity from ACC deaminase to D-cysteine desulfhydrase. Following a detailed phylogenetic study [13] of ACC deaminase genes, it was observed that ACC deaminase genes and D-cysteine desulfhydrase genes clustered separately, albeit nearby, to one another. Together, these data are consistent with, but do not prove, the possibility that ACC deaminase evolved from the enzyme D-cysteine desulfhydrase or a “similar pyridoxal phosphate dependent deaminase related to tryptophan synthase beta subunit and sharing a common origin” [13].

## 8. Plant Evolution

The founding event in the origin of plants and other photosynthetic eukaryotes was the acquisition of the chloroplast, the cellular organelle responsible for photosynthesis, and it is estimated to date to ~1.6 billion years ago. Chloroplasts originate from a cyanobacterium that was incorporated into a eukaryotic cell, most likely a freshwater protozoon that fed on bacteria, by phagocytosis and then transformed into an organelle to be inherited from one generation to the next. Molecular data support a monophyletic origin of chloroplasts in all plants [152,153,154].

The transformation of a phagocyted cyanobacterium was associated with a few changes in both organisms; the protozoon acquired a cell wall, lost the ability to feed by phagocytosis, and became autotrophic for carbon while the cyanobacterium became an organelle surrounded by two membranes. Both organisms underwent a rearrangement of their genomes, since a number of genes were transferred from the prokaryote to the nucleus of the protozoon. Following these events, the cyanobacterium became unable to live on its own, integrated into the eukaryotic cell, and the two eventually formed a permanent partnership. Most plant biologists agree on the above description concerning the origin of chloroplasts and plastid phylogenomic analysis has been used to explore the whole phylogeny of green plants [155,156]. However, it must be acknowledged that alternative hypotheses are still attempting to challenge the above-mentioned view [157].

A recent model for eukaryote classification is based on the organization into eight supergroups (plus some other smaller groups), which are rather different from the conventional eukaryotic Kingdoms: Amorphea (Amebozoa, Fungi, Animalia), Archaeplastida (Glaucophyta, Rhodophyta, and Chloroplastida), TSAR (Stramenopila, Alveolata, and Rhizaria), Haptista, Cryptista, CRuMs, Hemimastigophora, and Excavata [158]. Photosynthetic organisms are present in several supergroups, but here the focus is on Archaeplastida, since the photosynthetic members of other eukaryotic supergroups most likely originated through secondary symbiosis involving unicellular organisms belonging to the clade Archaeplastida.

Archeaeplastida comprise three main lines: Glaucophyta (a group of unicellular algae), Rhodophyta (red algae), and Chloroplastida [159]. In Glaucophyta and Rhodophyta, chloroplasts still morphologically resemble cyanobacteria, and their photosynthetic pigment composition is characterized by the presence of both chlorophyll a and phycobiliproteins, like cyanobacteria. Chloroplastida are green plants comprising “green algae” and land plants, with phycobiliproteins replaced with chlorophyll b, a change that allows for the reorganization and packing of the thylakoid membranes, the system found in chloroplasts and cyanobacteria [160,161,162]. In addition, green plants accumulate starch (the main carbon reserve material in Plantae) in chloroplasts, while Glaucophyta and Rhodophyta accumulate starch in the cytoplasm.

Green plants developed along two main lines, one was caused by the Chlorophyta, the other one by the Streptophyta. Chlorophyta represent the vast majority of green algae in the seas and freshwaters, while Streptophyta include a small group of freshwater algae, Charophytina, and all the land plants, Embryophytina. Streptophyta represent the most interesting group for the purposes of this review since, in this clade, ethylene production via the action of the enzymes ACS and ACO has been described in seed plants. 

Indeed, the ethylene reception system seems to be much older than that concerning the phytohormone biosynthesis since a gene, known as Etr1 or PIXA or UirS, coding for an ethylene receptor protein, was found in the cyanobacterium *Synechocystis*; the receptor comprises three domains: an ethylene binding one, a phytochrome-like one able to respond to photons, and a histidine kinase one, and it is involved in photaxis [163,164,165,166]. A recent study concerning the investigation of transcriptomic changes occurring in *Synechocystis* in response to ethylene during phototaxis revealed that the application of ethylene modulated over 500 gene transcripts [167]. Plants probably inherited this gene, through transfer from the plastid, about 1–1.5 billion years ago [42]. 

Data concerning chlorophyte green algae indicate that the ethylene receptor gene was lost when this group of organisms and the charophyte green algae diverged, over one billion years ago [168].

Ju et al. [62] investigated the origin of ethylene as a plant hormone by mining the transcriptome datasets of five species of charophytes, representative of the main lineages of these green algae, and performing a comparison with data from representatives of five lineages of embryophytes: mosses, lycophytes, gymnosperms, monocot angiosperms, and eudicot angiosperms. In the charophyte sequences (with some differences in the five considered species), homologues of all the central genes in the ethylene biosynthesis and the signaling pathway of plants were identified; *Spyrogyra pratensis* and *Choleochaete orbicularis* presented the most complete sets of ethylene-related homologous genes. Therefore, *S. pratensis* was tested for ethylene biosynthesis and response. The results showed that *S. pratensis* cultures produced detectable levels of ethylene and the levels of ethylene increased when exogenous ACC was provided to the medium but to a low extent compared with the amount of applied ACC. The enzyme possibly involved is yet unknown. The treatment of submerged *S. pratensis* cultures with exogenous ethylene resulted in dose-dependent cell elongation; this response was prevented using a prior treatment with a competitive inhibitor of ethylene binding in land plants. Ethylene-dependent cell elongation is a typical response to submersion in land plants [11]. It is worth mentioning that ACC was also able to induce cell elongation in the submerged cultures of *S. pratensis*, providing evidence that this alga can respond to endogenous ethylene. Finally, the transformation of the *Arabidopsis* ETR1 ethylene receptor, SpETR1, with the *S. pratensis* homologous gene was able to alleviate the short hypocotyl phenotype of the triple ethylene receptor mutant *Arabidopsis* etr1-7, etr2-3, and ein4-4. 

Overall, these results show that ethylene functions as a hormone in some lines of charophytes and their signaling pathways are conserved compared with *Arabidopsis*. Therefore, these results strongly suggest that ethylene was already a functional hormone in the common ancestor of charophytes and land plants, at least 450 million years ago.

Ethylene synthesis upon ACC treatment was observed in some additional organisms, like cyanobacteria, red algae, and some chlorophytes [169,170,171,172,173], but the enzymes for the synthesis of ethylene are unknown in these organisms.

An increasing body of evidence supports the idea that ACC plays a role as a signaling molecule on its own, beyond being the precursor of ethylene ([78] and Section 4 of this review). Both ACS activity and ACC were detected in liverworts, mosses, and ferns [172,173,174]. It is therefore possible that the system for ethylene perception may have evolved before the ability to synthesize ethylene from ACC, since the former was already present in cyanobacteria [167]. So far, ACO homologs have not been found in the genome sequences of non-seed plants [175], while ACS homologs are conserved in land plants, consistent with the notion that ACC biosynthesis evolved before the ACC-dependent synthesis of ethylene.

## 9. Concluding Remarks

Ethylene is one of the key phytohormones affecting a wide range of plant gene expression and subsequent behavior. The currently available data suggest that ethylene was already a functional plant hormone ~450 million years ago (around the time that land plants began to emerge). In addition, other data are consistent with the notion that ACC acted as a plant hormonal signal prior to the emergence of plant-produced ethylene as a phytohormone (with ACC currently retaining a vestige of that early hormonal activity). The enzyme, ACC deaminase, synthesized by numerous PGPB (which have positively interacted with plants for many millions of years), cleaves ACC thereby preventing it from being converted to ethylene and from acting as a phytohormone. It appears that ACC deaminase may have evolved from the enzyme D-cysteine desulfhydrase, or another similar pyridoxal phosphate-dependent deaminase related to the tryptophan synthase beta subunit. Taken together, these data help to facilitate an understanding of the central role played by ethylene in plant growth and development. Moreover, these data emphasize the key role of PGPB that produce ACC deaminase in regulating plant ACC and ethylene levels, especially during periods of stress.

## Figures and Tables

**Figure 1 biology-12-01043-f001:**
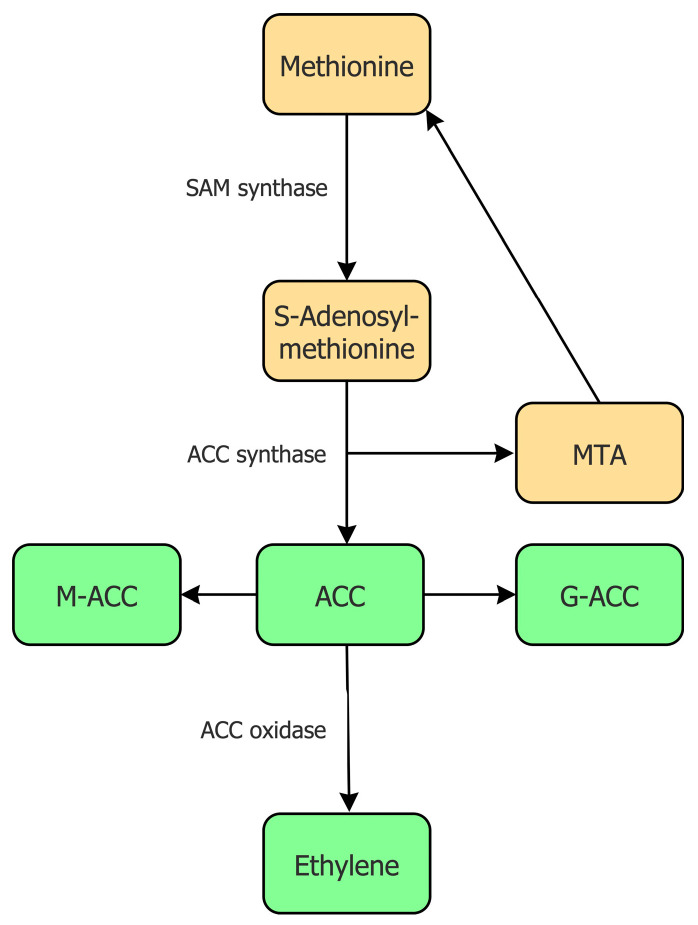
Overview of the biosynthetic pathway for the synthesis of ACC and ethylene in plants. Abbreviations: MTA, 5′-methylthioadenosine; M-ACC, 1-(malonyl)-ACC; G-ACC, 1-(glutamyl)-ACC. The enzymes catalyzing some of these reactions are shown to the left of the arrow indicating a catalyzed reaction.

**Figure 2 biology-12-01043-f002:**
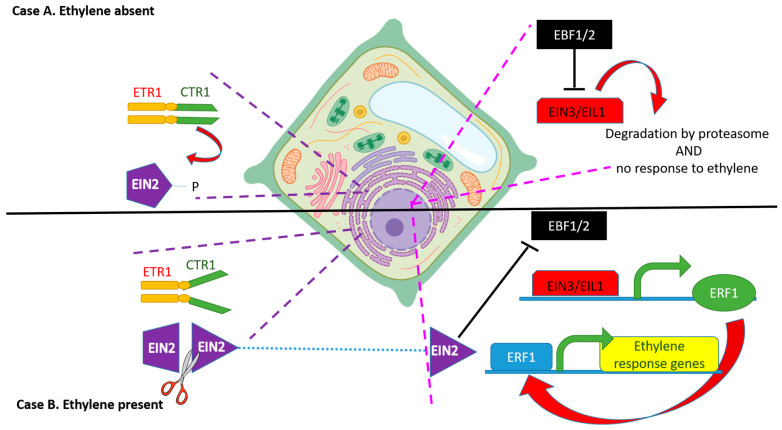
When ethylene is absent (Case A), the receptors located in the endoplasmic reticulum membrane (in the diagram this is represented by ETHYLENE RESPONSE 1, ETR1) repress downstream ethylene responses. A serine–threonine protein kinase called CONSTITUTIVE TRIPLE RESPONSE 1 (CTR1) phosphorylates EIN2 (ETHYLENE-INSENSITIVE 2) protein at the C terminal. In this way, EIN2 becomes targeted for degradation. In the nucleus, proteins EBF1 (EIN3 BINDING F-BOX1) and EBF2 (EIN3 BINDING F-BOX2) cooperate to activate the degradation of two transcription factors: EIN3 (ETHYLENE-INSENSITIVE 3) and EIL1 (ETHYLENE-INSENSITIVE 3–like 1). Altogether, these steps lead to inhibition of downstream ethylene signaling.

**Figure 3 biology-12-01043-f003:**
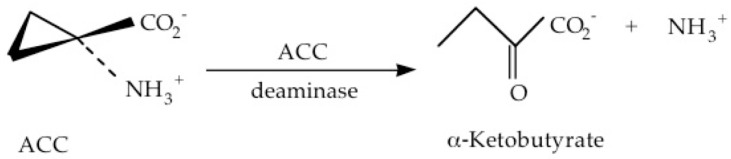
The cleavage of ACC by the enzyme ACC deaminase.

**Figure 4 biology-12-01043-f004:**
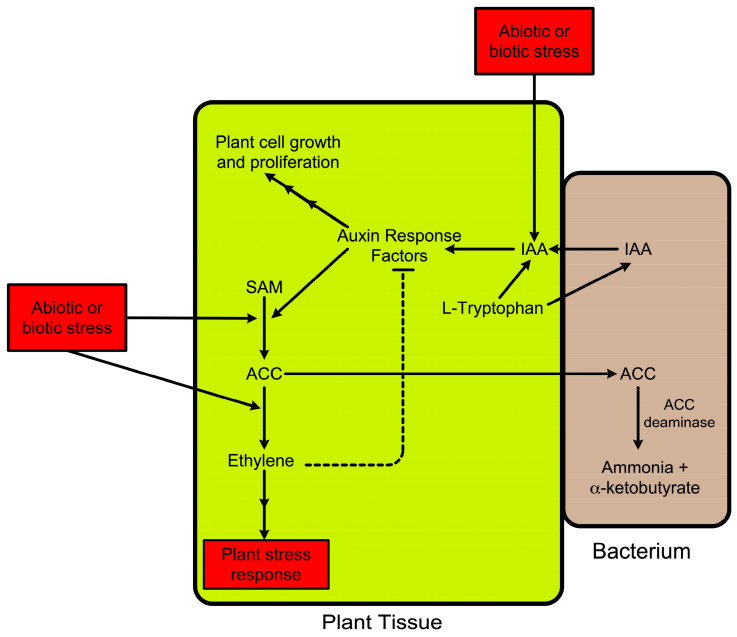
Model of ACC deaminase promoting plant growth.

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
