# Peer review of "Ethylene, ACC, and the Plant Growth-Promoting Enzyme ACC Deaminase"

_biology, 2023, doi:10.3390/biology12081043_

Round 1

Reviewer 1 Report (Previous Reviewer 1)

All my previous comments and suggestions have been addressed and revised. I suggest that this revised manuscript now is acceptable. 

Author Response

We thank reviewer 1 for this new report about our paper.

Reviewer 2 Report (Previous Reviewer 2)

There have been some improvements to the manuscript, but a few inaccuracies and contradictory sentences remain regarding the topics of evolution and mechanisms in ethylene signaling.

Lines 10-11 is misleading, because ACC is the immediate ethylene precursor only in seed plants. There is no evidence for ACC being the ethylene precursor in ferns, bryophytes, algae, etc. For example, there are studies with radiolabeled ACC showing that feeding non-seed plants with radiolabeled ACC does not lead to the production of radiolabeled ethylene. There is also no evidence of genes encoding ACC oxidase in non-seed plants (e.g., ferns, bryophytes, charophytes, etc) as the authors state on Lines 518-520.

Related to this, the Ju et al. Nature Plants (2015) paper on ethylene being a hormone in charophytes is incorrect about ethylene biosynthesis; this paper was published before it became commonly accepted that in non-seed plants, ACC is not converted to ethylene, on the basis of more and more genome sequences. This fact affects some of the misstatements in this manuscript, such as line 473 that claims ACS and ACO first appeared in plants in the Streptophyta. Also, Line 495 should point out that the amounts of ethylene produced were quite low relative to the amount of ACC applied to the alga. Assuming that this ethylene production occurred enzymatically, then the enzyme for this is actually unknown (alternatively, it could be due to non-specific enzymes or due to a chemical conversion of ACC to ethylene.) Otherwise, how do the authors reconcile this with their statement on Lines 518-520?

Line 19 is misleading because environmental ethylene (from bacteria, decaying matter, fires and volcanoes, etc) very likely predated the existence of ACC.

Line 189, Instead of saying “Alternatively” it would be more accurate to say,  “Additionally”. In fact, this additional function may be the predominant function of the EIN2 C-terminus. With respect to this point, it doesn’t seem helpful that in lieu of accuracy, the authors want to omit this in Figure 2 for the following reason: “in order to not complicate the figure, making it difficult to read”. I suggest that the authors at least state the correct process in the figure legend, so as not to disregard the findings that were published in two Cell papers (only 1 of which is cited on line 191). For example, on lines 212-213, this part “…inhibits the EBF1/EBF2 complex and modulates EIN3 and EIN3-Like (EIL1) transcription factors” could be modified to say “…inhibits translation of EBF1/EBF2 thus allowing accumulation of the EIN3 and EIN3-LIKE1 (EIL1) transcription factors ”.

Line 193-197: Where it is stated that “Finally, another role of EIN2 is the stabilization of two transcriptional factors…”, the authors need to know that this is actually the function of EIN2 binding to EBF1/EBF2 in the prior sentence (Lines 189-191), so saying “Finally, another role….” is incorrect. They are describing the result of what they stated on Lines 189-191.

Related to this, lines 191-194 are misplaced and should be switched with the sentence on lines 189-191, because Lines 193-197 are the details for Lines 189-191 and thus they need to be grouped together.

Lines 213-215 are misleading because EIN3 and EIL1 activate the transcription of many, many genes of the ethylene response. ERF1 is just one of those many genes. So it is incorrect to state that ERF1 alone activates the entire ethylene response, as stated on line 214..

Line 229: the “unknown but important function” could be for ethylene production, no?

Line 246, the authors could add “in seed plants” at the end of the sentence for clarity.

Line 379 is confusing, because ethylene is known to enhance seed germination in Arabidopsis.

Question: I’m not sure this review discussed why bacteria need the ACC deaminase enzyme? Does ACC/ethylene lead to plant defense mechanisms against the bacteria?

Optional suggestion. There are a lot of references (some redundant or obscure) that could be pared down. For example, reference 66 seems extraneous and unhelpful to cite, as I believe Ref 16 covers the topic accurately.

Small grammatical errors:

Line 17 change “is explored” to “are explored”

On Line 131, add an “s” to “molecule”

Lines 191-192, fix “This pathways”

The quality of English is very good. There are a few minor grammatical errors, such as the ones I pointed out, but there may be others I did not point out.

Author Response

There have been some improvements to the manuscript, but a few inaccuracies and contradictory sentences remain regarding the topics of evolution and mechanisms in ethylene signaling.

Lines 10-11 is misleading, because ACC is the immediate ethylene precursor only in seed plants. There is no evidence for ACC being the ethylene precursor in ferns, bryophytes, algae, etc. For example, there are studies with radiolabeled ACC showing that feeding non-seed plants with radiolabeled ACC does not lead to the production of radiolabeled ethylene. There is also no evidence of genes encoding ACC oxidase in non-seed plants (e.g., ferns, bryophytes, charophytes, etc) as the authors state on Lines 518-520.

The abstract line 10-11 has been modified accordingly.

Related to this, the Ju et al. Nature Plants (2015) paper on ethylene being a hormone in charophytes is incorrect about ethylene biosynthesis; this paper was published before it became commonly accepted that in non-seed plants, ACC is not converted to ethylene, on the basis of more and more genome sequences. This fact affects some of the misstatements in this manuscript, such as line 473 that claims ACS and ACO first appeared in plants in the Streptophyta. Also, Line 495 should point out that the amounts of ethylene produced were quite low relative to the amount of ACC applied to the alga. Assuming that this ethylene production occurred enzymatically, then the enzyme for this is actually unknown (alternatively, it could be due to non-specific enzymes or due to a chemical conversion of ACC to ethylene.) Otherwise, how do the authors reconcile this with their statement on Lines 518-520?

We thank the reviewer for this suggestion. We modified the text in order to avoid any kind of misunderstanding.

Line 19 is misleading because environmental ethylene (from bacteria, decaying matter, fires and volcanoes, etc) very likely predated the existence of ACC.

Lines 18 and 19 have been modified accordingly.

Line 189, Instead of saying “Alternatively” it would be more accurate to say,  “Additionally”. In fact, this additional function may be the predominant function of the EIN2 C-terminus.

Done.

With respect to this point, it doesn’t seem helpful that in lieu of accuracy, the authors want to omit this in Figure 2 for the following reason: “in order to not complicate the figure, making it difficult to read”. I suggest that the authors at least state the correct process in the figure legend, so as not to disregard the findings that were published in two Cell papers (only 1 of which is cited on line 191).

The figure legend has been modified accordingly.

For example, on lines 212-213, this part “…inhibits the EBF1/EBF2 complex and modulates EIN3 and EIN3-Like (EIL1) transcription factors” could be modified to say “…inhibits translation of EBF1/EBF2 thus allowing accumulation of the EIN3 and EIN3-LIKE1 (EIL1) transcription factors ”.

This sentence has been modified as required.

Line 193-197: Where it is stated that “Finally, another role of EIN2 is the stabilization of two transcriptional factors…”, the authors need to know that this is actually the function of EIN2 binding to EBF1/EBF2 in the prior sentence (Lines 189-191), so saying “Finally, another role….” is incorrect. They are describing the result of what they stated on Lines 189-191.

We modified the sentence to “Finally, EIN2 stabilize the two transcriptional factors EIN3 (ETHYLENE-INSENSITIVE 3) and EIL1 (ETHYLENE INSENSITIVE 3-like 1 protein…”

Related to this, lines 191-194 are misplaced and should be switched with the sentence on lines 189-191, because Lines 193-197 are the details for Lines 189-191 and thus they need to be grouped together.

We modified the order of the sentences.

Lines 213-215 are misleading because EIN3 and EIL1 activate the transcription of many, many genes of the ethylene response. ERF1 is just one of those many genes. So it is incorrect to state that ERF1 alone activates the entire ethylene response, as stated on line 214..

The sentence has been modified according to the reviewer’s suggestion.

Line 229: the “unknown but important function” could be for ethylene production, no?

This sentence has been deleted.

Line 246, the authors could add “in seed plants” at the end of the sentence for clarity.

The sentence has been modified accordingly.

Line 379 is confusing, because ethylene is known to enhance seed germination in Arabidopsis.

In the cited paper (Suarez et al., 2019) is clearly stated and demonstrated that “The loss of DAT1 leads to a significant increase of ethylene after D-Met application, resulting primarily in shortening of the hypocotyl and root in the dat1 mutants and Ler irrespective of the light regime……. The reciprocal accumulation of malonylated D-Met and ACC implies that the loss of AtDAT function or enzymatic activity results in over-accumulation of ACC that is then causing an increased ethylene production and eventually reduced seedling growth. However, this conclusion has to be reviewed critically, because even the dat1 mutant seedlings did not show the full spectrum of the canonical triple response, as tightening of the apical hook or thickening of the hypocotyl was only partially observed….. D-Met accumulation leads to an increase of ethylene concentrations, but possibly other compounds like ACC and its derivatives may also contribute to the observed physiological responses of dat1 affected plants. This would explain why the dat1 mutants do not show the full spectrum of triple response after treatment in the presence of D-Met. ”  

In conclusion, the Authors found an inspected behaviour, but hypothesized the involvement of ACC and other derivatives to explain the observed data.

Question: I’m not sure this review discussed why bacteria need the ACC deaminase enzyme? Does ACC/ethylene lead to plant defense mechanisms against the bacteria?

ACC deaminase has been shown to be important to both bacteria and plants during stressful environmental conditions. We think this is out of the topic covered by the review.

Optional suggestion. There are a lot of references (some redundant or obscure) that could be pared down. For example, reference 66 seems extraneous and unhelpful to cite, as I believe Ref 16 covers the topic accurately.

We are sorry but we don’t agree with the reviewer. In addition, one very recent reference suggested by another reviewer has been added to the text.

Small grammatical errors:

Line 17 change “is explored” to “are explored”

done

On Line 131, add an “s” to “molecule”

done

Lines 191-192, fix “This pathways”

done

Comments on the Quality of English Language

The quality of English is very good. There are a few minor grammatical errors, such as the ones I pointed out, but there may be others I did not point out.

We have addressed all grammatical errors that we were aware of.

Reviewer 3 Report (Previous Reviewer 3)

This revision has addressed my concerns. There is one new paper updating the model for the structure of the ethylene binding domain. This might be a good thing  to add.

Azhar  et al. (2023) Basis for high-affinity ethylene binding by the ethylene receptor ETR1 of Arabidopsis. Proceedings of the National Academy of Sciences, USA. 120: e2215195120.

Author Response

We thank the reviewer. The reference by Azhar  et al. (2023) has been added to the text

This manuscript is a resubmission of an earlier submission. The following is a list of the peer review reports and author responses from that submission.

Round 1

Reviewer 1 Report

These authors reviewed the roles of the phytohormone ethylene, ACC-ethylene’s immediate precursor and ACC deaminase produced by plant growth-promoting bacteria. ACC-deaminase-producing bacteria can facilitate plant growth and development through conversion of the immediate ethylene precursor ACC into α-ketobutyrate and ammonia, which can be further used by other associated microbes as substrates. There are currently a lot of studies on the positive effects of the plant growth-promoting enzyme ACC deaminase on plant growth and development, particularly in stressful environments. In the past decades, scientists have developed a much better understating of how bacterial ACC deaminase contribute to plant growth and stress tolerance. Nevertheless, it is interesting and innovative to investigate the roles of ACC and ACC deaminase from an evolutionary perspective. Overall, it is a well-written and well-constructed paper. I have only a few comments and suggestions as follows:

1. When these authors reviewed and discussed ACC as a signaling molecule, I think that there are some missing aspects on ACC may act as signaling molecules in plant-microbial interactions, which should be added to this section. ACC deaminase-producing bacteria could affect rhizosphere microbiome assembly via modulating plant ethylene/ACC levels, which subsequently may feedback on plant phenotypic traits (Chen et al., Microbiome, 2020; Kong and Liu, Frontiers in Plant Science, 2022).  

2. in the summary and abstract, a simple reason why these authors focused on the role of D-amino acids in this review should be added to make logic smoother. For instance, line26.

Reviewer 2 Report

I felt this review article has several problems.  

1. Unfortunately, the article is a collection of disjointed short summaries (with section headings such as “3. Ethylene as a signaling molecule”, “5. ACC deaminase”, “6. D-amino acids in plants”, “8. Plant evolution”). The topics in each section are not clearly connected to each other, and any overarching purpose in putting these topics together in this manuscript is only weakly conveyed. The authors could greatly improve on this by presenting deeper, better-sourced information that communicates coherent take-home messages.

2. The authors make some interesting statements that do not logically follow from the text they have written. For example, there is no discussion of ACC in primitive plants nor the evolutionary development of ethylene signaling to support the following sentence on lines 219-222: “Finally, assuming that ACC is in fact a plant signaling molecule, at least under certain circumstances, this implies that ACC may have been a major signaling molecule in primitive plants prior to the development of ethylene and ethylene signaling.” Similarly, the conclusion section sounds interesting, but the statements in the conclusion section don’t closely reflect the content presented in the manuscript. In short, clear arguments for interesting conclusions or statements appear to be mostly missing.

3. There is no evidence discussed to support this statement in the Summary: “It is likely that 1-aminocyclopropane-1-carboxylate was present prior to the existence of ethylene" (Lines 16-18). Similarly, lines 24-26 in the Abstract are not supported, “Given that ACC is a signaling molecule under some circumstances, this suggests that ACC, which evolved prior to ethylene, may have been a major signaling molecule in primitive plants”. Actually I think these statements are likely to be incorrect, because ethylene was present in the environment, and ethylene signaling machinery appears to have been present in algae, long before ACC is known to have been synthesized.

4. The pivotal topic that could help bind the article together, “4. ACC as a signaling molecule”, is very short and lacks in-depth descriptions and discussion. The authors cite a mixture of primary research papers and review articles to provide evidence of ACC functioning as a signal. However, at least two recent primary research papers are absent and these examples are not included within the cited review articles: Li et al. (2020) doi: 10.1038/s41477-020-00784-y and Mou et al. (2020) doi: 10.1038/s41467-020-17819-9.

5. The authors’ model on the role of ACC deaminase in promoting plant growth (lines 279-293 and Figure 4) is unclear. After ACC is exuded from the plant cell (stated on line 280), wouldn’t this ACC no longer be available to be enzymatically converted into ethylene? If so, then whether this exuded ACC is taken up by bacteria and cleaved by bacterial ACC deaminase in the bacterial cell seems irrelevant to ethylene production.

6. Figure 2 shows a nucleus and ER, but this is confusing because the components of ethylene signaling are incorrectly shown in relation to these cell elements. The figure implies that the ethylene receptors and EIN2 are cytosolic. It is known that the N-terminal domain of the receptors and EIN2 lie within the ER membrane. EIN3/EIL1 and ERF1 are nuclear.  

7. Lines 12 and 21: I think it needs to be stated that the ACC deaminase enzyme is a bacterial enzyme, e.g., “how the bacterial enzyme ACC deaminase cleaves plant-produced ACC….”

8. There are also some inaccurate statements and missing information, especially regarding ethylene signaling. Below are some corrections and suggestions mainly on the ethylene signaling section:

a. Lines 9 and 10. ACC is NOT the immediate precursor of ethylene in all plants. ACC is the immediate precursor of ethylene in angiosperms (with some exceptions), and it is likely the ethylene precursor in gymnosperms. Is it not the ethylene precursor in ferns, liverworts, mosses, etc, which lack ACC oxidase (which converts ACC to ethylene) based on genome evidence and experiments with radiolabeled ACC (see original papers cited in a review article by Li et al., Curr. Opin. in Plant Biol, 2022, 65:102116)

b. Lines 152-153. RAN1 is NOT a “copper-based cofactor”. RAN1 is a copper transporter. The sentence would be correct if re-worded as follows: “The binding of ethylene with its receptor is supported by a copper-based cofactor, which is required for ethylene receptor function and provided by the RAN1 (Responsive to Antagonist1) copper transporter.”

c. A reference needs to be cited for the point made on lines 111-115.

d. The wording of sentence on lines 158-160 is unclear. I agree that the structure of the ethylene binding domain has been unknown. However, the sentence seems to be saying that the functions of the ETR1 ethylene receptor and its homologs were unknown for a long time, yet the functions of these receptors were identified 25-30 years ago.

e. Line 170: Change “by ethylene receptor 1” to “by the ethylene receptors”

f. Line 172: EIN2 is predicted to have 12 N-terminal transmembrane domains. Definitely not 20!

g. Lines 175-185 omit two papers published in Cell on the function of EIN2 in P-bodies (reviewed in Ref  16. Binder, 2020). This step is also missing in figure 2.

h. Line 176: Capitalize “Ethylene Insensitive 3” at this first mention, and remove the spelled-out description of EIN3 on lines 181 and 193. Same with EIL1 in the figure legend.

i. Line 177: Change “when ethylene is synthesized by the plant” to “when ethylene is perceived by the plant”.

j. Lines 179-180: Change “the site of perception to the ER membrane” to “the site of perception at the ER membrane”

k. Line 182 Spell out EIL1 properly (EIN3-LIKE 1). Also, change “homologous of EIN3” to either “homologous to EIN3” or “homolog of EIN3”

l. Line 197: “EFB1/EFB2” should be “EBF1/EBF1”

m. Lines 197-198: “indirectly triggers EIN3” is inaccurate. Please check the Binder (2020) reference and the primary literature. For example, EIN3/EIL1 proteins accumulate when they are not targeted for turnover by EBF1/EBF2.

n. Line 203 and other places in the manuscript: I think it is premature to refer to ACC as a “plant hormone”. ACC is not considered to be a plant hormone. It would be better to say that ACC appears to function as a plant signal.

o. Line 326: should “other two families” be changed to “two other families”?

Reviewer 3 Report

In general, this is a well-written and interesting review. I like the speculation that ACC and ethylene signaling initially evolved separately. My main critique with the review is that it seems rather selective ignoring more recent information that impacts their conclusions. The predominant concern is that you seem to have ignored papers from the past 8 years or so showing that ethylene signaling predates plants. This seems an oversight given the evolutionary focus of their review.

You claim that ACC evolved prior to ethylene. This actually seems highly unlikely since ethylene is produced abiotically (and thus was almost certainly present early in the formation of the planet) and ethylene is also produced by mechanisms independent of ACC. Additionally, putative ethylene receptors in non-pathogenic bacteria are wide-spread across species including alpha, beta, gamma proteobacteria, planctomycetes, bacterioidetes and verrucomicrobia. And, at least one functional ethylene receptor has been shown to exist in the cyanobacteria Synechocystis from work in the Bleecker and Binder labs. See below comment for more on this. There’s a review by Carlew et al. if more information is needed about this.  Thus, the claimed evolutionary timing is not supported. It’s too bad this is not also discussed since I think that this provides additional support for ethylene and ACC independently evolving as signaling molecules

Lines 445-446: This statement is very out of date. There are several papers from the Binder lab showing that Synechocystis contains a functional ethylene receptor that affects phototaxis, motility, type IV pili formation, and other responses. And work of this lab along with papers from several other labs (Ramakrishan and Tabor 2016; Song et al. 2011; Narikawa et al. 2011) has delineated the signaling pathway. Although, different from the plant pathway, these studies show that signaling occurs. Interestingly, this Synechocystis ethylene receptor is also a photoreceptor. There is an additional paper showing that the filamentous cyanobacterium Geitlerinema also responds to ethylene.

Minor points:

line 121: Ethylene isn’t the simplest signaling molecule with hormone-like behavior in plants. This would be more appropriately applied to NO with only 2 atoms.

line 136: We can’t know if “all the molecular elements” have been identified. Perhaps change to “All the major molecular elements”.

lines 152-154: This statement needs to be corrected. RAN1 is not a copper-based co-factor. It is a copper transporter that delivers copper to the lumen of the ER.

lines 161-164: While this was hypothesized in the 1980’s in this paper, there is not really any support for this idea.

line 172: EIN2 has 12 (not 20) transmembrane domains.

line 203: Tsuchisaka et al 2009 in a paper from the Theologis lab provided early evidence that ACC is a signaling molecule. This citation needs to be added. Given that this study focused on ACC synthase isoforms in plants, it may provide more ideas for this review.

lines 491-493:  As with my critiques above, this statement is misleading. Yes, ACC likely was a signaling molecule prior to land plants, but so was ethylene.
